# Pre-drinking, alcohol consumption and related harms amongst Brazilian and British university students

**Mariana G. R. Santos**[1], **Zila M. Sanchez**[1], **Karen Hughes**[2], **Ivan Gee**[3], **Zara Quigg**[3]*

**1** Department of Preventive Medicine, Universidade Federal de São Paulo, São Paulo, Brazil, **2** School of Human Sciences, Bangor University, Wrexham, United Kingdom, **3** Public Health Institute, Liverpool John Moores University, Liverpool, United Kingdom

* z.a.quigg@ljmu.ac.uk

**Data Availability Statement:** All relevant data are within the paper and its Supporting information files.

## Abstract

Drinking in private or other unlicensed settings before going out (i.e., pre-drinking) is increasingly being identified as a common behaviour amongst students as it provides an opportunity to extend their drinking duration and socialise. However, studies suggest associations between pre-drinking and alcohol-related harms. This study examines Brazilian and British university students' pre-drinking patterns and associations with nightlife-related harms amongst drinkers. A total of 1,151 Brazilian and 424 British students (aged 18+ years) completed an online survey. The questionnaire covered sociodemographic variables, nightlife drinking behaviour including pre-drinking and past 12 months experience of alcohol-related harms. Most participants were female (BRA 59.1%, ENG 65.3%; $p = 0.027$), undergraduate students (BRA 88.2%, ENG 71.2%; $p<0.001$) and aged 18–25 years (BRA 78.8%, ENG 81.5%; $p<0.001$). Pre-drinking was more prevalent in England (82.8%) than Brazil (44.0%; $p<0.001$), yet Brazilian students drank more units of alcohol than British students when pre-drinking (BRA 17.6, ENG 12.1; $p<0.001$). In multi-variate analyses, pre-drinking was significantly associated with increased odds of experiencing a range of harms across both countries (e.g., blackouts; failing to attend university), with the strength of associations varying between countries. Pre-drinking in Brazil and in England is an important event before going out amongst university students, however our study shows it is associated with a range of harms in both countries. Thus, preventing pre-drinking may be a crucial strategy to reduce excessive alcohol consumption and related harms in the nightlife context across countries with diverse nightlife environments and alcohol drinking cultures.

## Introduction

Harmful drinking has been recognised globally as a major public health issue, since drinking a higher quantity and more regularly increases the risks of a wide range of health conditions and social harms [1, 2]. In many countries, much of the burden of alcohol on health and crime is related to harmful drinking amongst young adults [3]. Research on young adults' drinking

**Funding:** MGRS. Doctoral fellowship funded by the Brazilian Government agency CNPq - National Council of Scientific and Technologic Development (process GDE 232375/2014-3). The sponsorship agency had no role in study design, data collection and analysis, decision to publish, or preparation of the manuscript.

**Competing interests:** The authors have declared that no competing interests exist.

behaviour has revealed that despite their knowledge about the negative consequences of drinking excessively, they are still motivated to drink for pleasure, be sociable, meet new people, feel good, and enjoy the state of drunkenness [4–6]. Thus, much harmful drinking among young adults occurs in the context of nightlife, including the phenomenon of pre-drinking; the consumption of alcohol in private or unlicensed settings prior to attending bars or nightclubs (also known as pre-loading, pre-partying, or pre-gaming [7–9].

Pre-drinking has been found to be common among young people in many countries, often motivated by reasons such as saving money on alcohol during a night out, reducing social anxiety and the desire for drunkenness [10–12]. However, pre-drinking has also been associated with higher alcohol use, greater drunkenness, and a range of harms such as violence in nightlife settings [13, 14]. Studies suggest that drinkers tend to drink almost twice as much on pre-drinking evenings compared to other drinking evenings [15–17], and they usually drink more often and in greater quantities per occasion, compared to non-pre-drinkers [11, 18, 19].

Most studies on pre-loading have been conducted in European or North American settings [20, 21], with research in other cultures being scarce. However, studies from Brazil have also found pre-drinking to be common among young adults and to be associated with rapid alcohol consumption and an increased risk of harms such as blackouts, alcohol poisoning, risky sexual behaviours, and violence [22–24]. One study found that pre-drinkers' main motive was to save money on drinks, yet that pre-drinkers actually drank more alcohol whilst in bars and nightclubs when compared with non-pre-drinkers [24]. However, more research is needed to understand the pre-drinking phenomenon in Brazil where despite a strong nightlife culture, research on young people's drinking patterns within nightlife and related risks is still in its infancy. Further, efforts to introduce public policies on alcohol control in Brazil have not yet been successful and there is no well-established prevention activity in place. Thus, developing understanding on pre-drinking practices in Brazil and how these may differ from those in other countries where alcohol control policies and prevention activity are better established can be beneficial in informing the development of alcohol policy and practice.

To date, few countries have explored differences in pre-drinking behaviours, motivations, and associated harms between young people from different cultures. Specifically, to our knowledge no previous studies have compared pre-drinking behaviours amongst university students; despite the university period often increasing harmful drinking amongst young people [25, 26]. Considering that university students attending nightlife and pre-drinking events in Brazil and England have both been found to be at greater risk of alcohol intoxication levels [24, 27–29], this study sought to examine and compare pre-drinking practices amongst Brazilian and English university students.

## Methods

A cross-sectional online survey recruited undergraduate and postgraduate students aged 18 years plus enrolled at Liverpool John Moores University (LJMU) in the UK and at Federal University of São Paulo (UNIFESP) in Brazil between March and July 2017. The study was approved by both institutions' ethics committees (UNIFESP—protocol number 1.845.314 CAAE: 61290216.3.0000.5505 and LJMU—16/CPH/005).

### Study design and sample

The inclusion criteria were: 1) being enrolled at LJMU or UNIFESP; 2) being at least 18 years old; and 3) alcohol consuming nightlife users. Students were approached via e-mail. School directors from each institution received an e-mail introducing the study and asking them to act as gatekeepers. They were informed that this would involve them sending an e-mail to

students that introduced the study and sought their voluntary and confidential participation, with a link to the online participant information sheet and questionnaire. The participants provided electronic consent prior to completing the questionnaire and were informed that they could withdraw from the survey at any time up to the point of submission. An estimated 12,896 e-mails were sent to UNIFESP students and 860 to LJMU students.

To boost participation, the self-reported questionnaire was also widely disseminated through social media platforms such as Facebook and Twitter, amongst students who participated in LJMU and UNIFESP university online groups. Using the sample size calculator by Raosoft, Inc [30], based on a 50% rate of population sample, 5% margin of error and 95%CI, the estimated sample needed in the two locations was 378 LJMU students and 375 UNIFESP students. In Brazil, of 1,491 that completed the questionnaire, 340 (22.8%) were screened out due to reporting that they never drink alcohol, resulting a final sample of 1,151 students. In England, of 493 that completed the questionnaire, 69 (14.0%) were screened out, resulting in a final sample of 424 students. In both institutions the calculated sample size was reached.

## Instrument and variables

The instrument used was designed based on previous surveys exploring alcohol consumption and related harms in UK nightlife settings [14, 31, 32], then reviewed and adapted to each country. A convenience sample of English and Brazilian students was recruited through informal networks to test the developed questionnaire. This pilot study was conducted in February 2017. The instrument was produced in both the English and Portuguese languages and covered sociodemographic characteristics, pre-drinking characteristics, nightlife drinking patterns and past 12 months experience of alcohol-related harms in the nightlife context. A screening question was used to identify those who had consumed alcohol [*"How often do you drink alcohol?"*, with the options of *'Never'*; *'Monthly or less'*; *'2 to 3 times a month'*; *'Once a week'*; *'2–4 days a week'* and *'5 or more times a week'*] in which those who answered *'Never'* were screened out of the survey and directed to the end of it.

Sociodemographic characteristics included age, gender, marital status, ethnicity, and academic year. Pre-drinking was identified by the response to *"Would you normally pre-drink before going out?"*, with options of *'Yes'* and *'No'*. It also included questions regarding pre-drinking characteristics such as: place for practice [*"Where would you normally pre-drink?"*, with options of *'Own home'*; *'Friend's home'*; *'Outside (e.g., park, beach)'* and *'Other'*]; main motivation [*"What is your main reason for pre-drinking?"*, with options of *'Part of going out'*; *'To socialise'*; *'To save money'*; *'To not go out sober'*; *'To lose control'*; *'To deliberately get drunk'*; *'To increase confidence'*; *'To relax'*; *'To feel like part of a group'*; *'To have fun'*; *'To increase mood'*; *'To reduce anxiety'* and *'Other motive'*]; and, food consumption [*"Would you typically consume any food during pre-drinking events?"*, with the options of *'Yes (snacks)'*, *'Yes (a meal)'* and *'No'*]. To investigate students' alcohol use during a night out, participants were asked *"How many of each of these drinks—Spirits, Wine, Beer and Alcopop—would you have a) during pre-drinking event? and b) at nightclubs, bars, and pubs?"*, with the options of *'0'* through to *'11 +'*. Participants were also asked if they have experienced selected alcohol-related risky behaviours in nightlife in the past 12 months (see list in Table 5).

## Statistical analysis

Data was analysed using IBM SPSS Statistics 24. To facilitate interpretation of results and have more precise estimates some similar categories with low frequencies were grouped and missing values excluded.

Brazil and England have many differences regarding type of drinks and glass/bottle sizes, and because the Brazilian government does not have an official definition of a unit of alcohol, the variables related to quantity of alcohol were recoded using the UK definition, according to which one unit is equal to 10 ml or 8 g of pure alcohol [33]. To calculate the alcohol amount consumed between countries, drinks were coded into standard UK units by multiplying the total volume of an alcoholic drink (ml) by its alcohol content (using its ABV measure–alcohol by volume) and dividing the result by 1,000 (Table 1).

The total quantity of alcohol consumed during a night out variable was formed by the sum of total amount of each type of drinks, such as: total amount of spirits (e.g., those who reported consuming bottle, single or double measures were grouped); total amount of wine (e.g., those who reported consuming bottle, small, standard, or large glasses were grouped); total amount of beer (e.g., those who reported consuming bottle, can, pin or half pint were grouped); and, total amount of alcopop (e.g., those who reported consuming large, standard or can were grouped).

To examine the differences between the two countries in students' pre-drinking behaviour; students' alcohol consumption and students' alcohol-related harms, frequency tables and descriptive statistics were computed and explored using Chi-Square tests. For the continuous variables that had a non-normal distribution–quantity of alcohol consumed within nightlife context–data were analysed by using Mann-Whitney U test to explore the difference in medians between groups.

Multivariate logistic regression (enter method) was performed and split by country (Brazil and England) to further explore the differences between them. The first analysis explored factors associated with students' pre-drinking. Pre-drinking was used as the dependent variable and the independent variables were age, gender, marital status, ethnicity, and academic year. The second analysis explored students' pre-drinking behaviour as a risk factor for alcohol-

**Table 1. Brazil and British drinks measures included in the questionnaires.**

|  | Country | | | |
|---|---|---|---|---|
|  | England | | Brazil | |
|  | (ml) | Units | (ml) | Units |
| **Spirits (ABV 37.5%– 40%)** |  |  |  |  |
| Bottle | 700 | 26 | 1000/750 | 37.5/30 |
| Single measure (standard) | 25 | 1 | 40 | 1.6 |
| Double measure | 50 | 2 | - | - |
| **Wine (ABV 12%)** |  |  |  |  |
| Bottle | 750 | 9 | 750 | 9 |
| Small glass | 125 | 1.5 | - | - |
| Standard glass | 175 | 2.1 | 150 | 1.8 |
| Large glass | 250 | 3.0 | - | - |
| **Beer/cider (3.6%–5%)** |  |  |  |  |
| Bottle | 330 | 1.7 | 600/355 | 3/1.7 |
| Can | 440 | 2.0 | 350/300 | 1.7/1.5 |
| Pint of regular beer/cider | 568 | 2.0 | - | - |
| ½ pint of regular beer/cider | 284 | 1.1 | - | - |
| **Alcopop (ABV 5.5%)** |  |  |  |  |
| Large bottle | 700 | 2.8 | - | - |
| Standard bottle | 275 | 1.5 | 275 | 1.5 |
| Can | 250 | 1.3 | - | - |

related harms amongst drinkers, using pre-drinking as an independent variable, controlled by sociodemographic variables (age, gender, marital status, ethnicity, academic year, and quantity of alcohol consumed within nightclubs, bars, and pubs settings) and each risky behaviour evaluated as dependent variables. The quantity of alcohol consumption in nightlife settings was included in the model as a control variable because the analysis aimed to investigate how pre-drinking behaviour would affect alcohol-related harms, independent of alcohol consumption occurring in nightlife settings.

## Results

Most participants were aged 18–25 years (BRA 78.8%, ENG 81.5%; $p < 0.001$); female (BRA: 59.1%, ENG: 65.3%; $p = 0.027$); self-categorised as being of white ethnicity (BRA: 71.4%, ENG: 89.4%; $p < 0.001$) and undergraduate students (BRA: 88.2%, ENG: 71.2%; $p < 0.001$). Regarding marital status, 59.4% of Brazilian students were single compared with 45.5% of British students ($p < 0.001$). There were statistically significant differences between Brazilian and British pre-drinkers regarding to age ($p < 0.001$), gender ($p = 0.001$), marital status ($p < 0.001$), ethnic group ($p < 0.001$) and academic year ($p < 0.001$) (Table 2).

The majority (82.8%) of British students reported pre-drinking, compared with 44.0% of Brazilian students ($p < 0.001$). Amongst pre-drinkers, in Brazil, the most common setting for practicing pre-drinking was at a friend's home (44.6%, compared with 45.3% in England) whilst in England it was in their own home (54.1%, compared with 24.3% in Brazil). In Brazil, 29.6% of students reported practicing pre-drinking outside (e.g., park, beach) compared with 0.0% in England ($p < 0.001$). Fewer British students reported pre-drinking to save money (44.7%) than Brazilian university students (64.9%) ($p < 0.001$). Regarding mixing food with drinking during pre-drinking, 80.4% of Brazilian students reported consuming food whilst pre-drinking, compared with 69.4% of British students ($p < 0.001$) (Table 3). Regarding the

**Table 2. Distribution of sociodemographic variables according to pre-drinking practice amongst Brazilian and British university students.**

| | Pre-drinking "Yes" | | | | | Pre-drinking "No" | | | | | Total | | | | |
|---|---|---|---|---|---|---|---|---|---|---|---|---|---|---|---|
| | Brazil N = 507 | | England N = 351 | | | Brazil N = 644 | | England N = 73 | | | Brazil N = 1,151 | | England N = 424 | | |
| | N | % | N | % | p value | N | % | N | % | p value | N | % | N | % | p value |
| **Age (years)** | | | | | <0.001 | | | | | <0.001 | | | | | <0.001 |
| 18–21 | 212 | 41.8 | 223 | 63.5 | | 238 | 37.0 | 14 | 19.2 | | 450 | 39.1 | 237 | 55.9 | |
| 22–25 | 200 | 39.4 | 85 | 24.2 | | 257 | 39.9 | 23 | 31.5 | | 457 | 39.7 | 108 | 25.5 | |
| 26+ | 95 | 18.7 | 43 | 12.3 | | 149 | 23.1 | 36 | 49.3 | | 244 | 21.2 | 79 | 18.6 | |
| **Gender** | | | | | 0.001 | | | | | 0.968 | | | | | 0.027 |
| Female | 272 | 54.0 | 226 | 65.7 | | 406 | 63.1 | 45 | 63.4 | | 678 | 59.1 | 271 | 65.3 | |
| Male | 232 | 46.0 | 118 | 34.3 | | 237 | 36.9 | 26 | 36.6 | | 469 | 40.9 | 144 | 34.7 | |
| **Marital status** | | | | | <0.001 | | | | | 0.030 | | | | | <0.001 |
| Single | 342 | 67.5 | 164 | 46.7 | | 342 | 53.1 | 29 | 39.7 | | 684 | 59.4 | 193 | 45.5 | |
| In a relationship | 165 | 32.5 | 187 | 53.3 | | 302 | 46.9 | 44 | 60.3 | | 467 | 40.6 | 231 | 54.5 | |
| **Ethnic group** | | | | | <0.001 | | | | | 0.006 | | | | | <0.001 |
| White | 364 | 71.8 | 316 | 90.0 | | 458 | 71.1 | 63 | 86.3 | | 822 | 71.4 | 379 | 89.4 | |
| Other | 143 | 28.2 | 35 | 10.0 | | 186 | 28.9 | 10 | 13.7 | | 329 | 28.6 | 45 | 10.6 | |
| **Academic year** | | | | | <0.001 | | | | | <0.001 | | | | | <0.001 |
| Undergraduate | 455 | 89.7 | 278 | 79.2 | | 560 | 87.0 | 24 | 32.9 | | 1015 | 88.2 | 302 | 71.2 | |
| Post-graduate | 52 | 10.3 | 73 | 20.8 | | 84 | 13.0 | 49 | 67.1 | | 136 | 11.8 | 122 | 28.8 | |

**Table 3. Pre-drinking characteristics amongst Brazilian and British university students.**

| | Brazil | | England | | |
| | N = 1,151 | | N = 424 | | |
| | N | % | N | % | *p* value |
|---|---|---|---|---|---|
| **Pre-drinking practice** | | | | | <**0.001** |
| Yes | 507 | 44.0 | 351 | 82.8 | |
| *Of pre-drinkers* | N = 507 | | N = 351 | | |
| **Pre-drinking place** | | | | | <**0.001** |
| At home | 123 | 24.3 | 190 | 54.1 | |
| At a friend's home | 226 | 44.6 | 159 | 45.3 | |
| Outside (e.g., park, beach) | 150 | 29.6 | 0 | 0.0 | |
| Other local | 8 | 1.6 | 2 | 0.6 | |
| **Pre-drinking main reason** | | | | | <**0.001** |
| Part of going out | 19 | 3.7 | 35 | 10.0 | |
| To socialize | 57 | 11.2 | 55 | 15.7 | |
| | Brazil | | England | | |
| | N = 507 | | N = 351 | | |
| | N | % | N | % | *p* value |
| **Pre-drinking main reason** | | | | | |
| To save money | 329 | 64.9 | 157 | 44.7 | |
| To not go out sober | 23 | 4.5 | 37 | 10.5 | |
| To lose control | 3 | 0.6 | 0 | 0.0 | |
| To get drunk | 10 | 2.0 | 9 | 2.6 | |
| To increase confidence | 3 | 0.6 | 8 | 2.3 | |
| To relax | 11 | 2.2 | 5 | 1.4 | |
| To feel part of a group | 5 | 1.0 | 2 | 0.6 | |
| To have fun | 34 | 6.7 | 22 | 6.3 | |
| To increase mood | 4 | 0.8 | 7 | 2.0 | |
| To reduce anxiety | 8 | 1.6 | 11 | 3.1 | |
| Other motive | 1 | 0.2 | 3 | 0.9 | |
| **Food consumption whilst pre-drinking** | | | | | <**0.001** |
| Yes | 408 | 80.5 | 244 | 69.5 | |

median number of total alcohol units (i.e., reported drinking any alcohol), amongst pre-drinkers, Brazilian students reported drinking a median of 17.5 units of alcohol compared with 12.1 units for British students U = 70996.0, *p*<0.001). Amongst non-pre-drinkers, Brazilian students reported drinking a median of 16.6 units of alcohol on on-licensed premises compared with 8.2 units for British students (U = 13317.5, *p*<0.001).

Table 4 shows the logistic regression results that identify factors associated with pre-drinking. In Brazil, male and single students had greater odds of pre-drinking (Odds Ratio [OR] 1.42; and, OR 1.74, respectively). In England, younger and undergraduate students had greater odds of pre-drinking (aged 18–21 years OR 5.00, aged 22–25 years OR 2.52; and, OR 3.84, respectively).

There were significant differences between the samples regarding the median number of alcohol units consumed at nightclubs, bars, and pubs. Of total alcohol consumed whilst at 7nightclubs, bars and pubs, Brazilian students typically reported drinking a median of 17.4 units of alcohol compared with 9.0 units for British students (U = 52780.5, *p*<0.001).

Regarding experiencing any kind of alcohol related harms in the last 12 months after attending nightclubs, bars and pubs, a greater proportion of British students reported

**Table 4. Factors associated with pre-drinking amongst Brazilian and British university students.**

| | Brazil | | | England | | |
|---|---|---|---|---|---|---|
| | N = 1,151 | | | N = 424 | | |
| | OR | 95% CI | *p* value | OR | 95% CI | *p* value |
| **Age (years)** | | | | | | |
| 18–21 | 1.26 | [0.88, 1.78] | 0.193 | 5.00 | [2.06, 12.12] | **<0.001** |
| 22–25 | 1.14 | [0.82, 1.60] | 0.414 | 2.52 | [1.27, 4.96] | **0.008** |
| 26+ (ref) | 1.00 | - | - | 1.00 | - | - |
| **Gender** | | | | | | |
| Male | 1.42 | [1.12, 1.81] | **0.004** | 0.92 | [0.50, 1.71] | 0.810 |
| Female (ref) | 1.00 | - | - | 1.00 | - | - |
| **Marital status** | | | | | | |
| Single | 1.74 | [1.36, 2.23] | **<0.001** | 1.05 | [0.58, 1.90] | 0.870 |
| In a relationship (ref) | 1.00 | - | - | 1.00 | - | - |
| **Ethnic group** | | | | | | |
| White | 1.10 | [0.84, 1.43] | 0.463 | 0.85 | [0.36, 2.01] | 0.715 |
| Other (ref) | 1.00 | - | - | 1.00 | - | - |
| **Academic year** | | | | | | |
| Undergraduate | 1.13 | [0.75, 1.70] | 0.542 | 3.84 | [1.86, 7.94] | **<0.001** |
| Post-graduate (ref) | 1.00 | - | - | 1.00 | - | - |

Note: reference for categories for each variable are identified with (ref).

experiencing physical violence (ENG 15.2%, BRA 6.2%, *p*<0.001), sexual harassment (ENG 22.7%, BRA 17.6%, *p* = 0.023) and practicing unprotected sex (ENG 20.4%, BRA 15.2%, *p* = 0.015). In England, 52.8% of the students reported experiencing blackouts, vomiting or coma, compared with 41.1% of the students in Brazil (*p*<0.001). Waking up feeling embarrassed about things done on the night before (ENG 47.4%, BRA 32.6%, *p*<0.001), being refused entry to another nightlife venue because of being too drunk (ENG 16.8%, BRA 1.9%, *p*<0.001), spoiling someone else's night out because of drinking (ENG 19.2%, BRA 13.6%, *p* = 0.006), failing to attend at university (ENG 42.2% BRA, 15.8%, *p*<0.001) and missing work because of drinking (ENG 8.3% BRA 4.5%, *p* = 0.004) were also more frequent in England than in Brazil (Table 5).

Regarding pre-drinking as a risk factor for nightlife-related harms (amongst drinkers), in Brazil pre-drinkers had 3.65 greater odds, compared to non-pre-drinkers, of reporting any kind of road traffic accident, 2.42 greater odds of reporting any kind of physical violence, 1.91 greater odds of having unprotected sex after going to nightclubs, bars and pubs, and 3.01 greater odds of regretting a decision to engage in sexual activity. Moreover, Brazilian pre-drinkers had 2.3 greater odds of falling asleep somewhere inappropriate after going to nightclubs, bars, and pubs; 7.07 greater odds of being refused entry to a venue for being too drunk; 1.94 greater odds of spoiling someone's night out for being too drunk, and 2.38 greater odds of missing work because of drinking. As for experiencing blackouts, vomiting or coma after going to nightclubs, bars, and pubs, in Brazil pre-drinkers had 2.18 greater odds of reporting such experience, whereas in England pre-drinkers had 3.86 greater odds. In Brazil pre-drinkers had 1.84 greater odds of waking up feeling embarrassed about things done on the night before, whereas in England pre-drinkers had 2.02 greater odds. Moreover, in Brazil pre-drinkers had 2.91 greater odds of failing to attend at university because of drinking, whereas in England pre-drinkers had 4.17 greater odds (Table 6).

**Table 5. Brazilian and British students' past 12 months alcohol-related harms experiences after attending nightclubs, bars, and pubs.**

| | After attending nightclubs, bars, and pubs: | | | | |
| --- | --- | --- | --- | --- | --- |
| | In Brazil | | In England | | |
| | N = 1,105 | | N = 422 | | |
| | N | % | N | % | *p* value |
| **Experienced any kind of road traffic accident** | 36 | 3.3 | 15 | 3.6 | 0.773 |
| **Experienced any kind of physical violence** | 68 | 6.2 | 64 | 15.2 | **<0.001** |
| **Experienced any kind of sexual harassment** | 195 | 17.6 | 96 | 22.7 | **0.023** |
| **Had unprotected sex** | 168 | 15.2 | 86 | 20.4 | **0.015** |
| **Regretted a decision to engage in sexual activity** | 106 | 9.6 | 54 | 12.8 | 0.068 |
| **Experienced blackouts, vomiting or coma** | 457 | 41.4 | 223 | 52.8 | **<0.001** |
| **Fallen asleep somewhere inappropriate** | 152 | 13.8 | 61 | 14.5 | 0.724 |
| **Woke up feeling embarrassed** | 360 | 32.6 | 200 | 47.4 | **<0.001** |
| **Were refused entry to a nightclub, bar, or pub** | 21 | 1.9 | 71 | 16.8 | **<0.001** |
| **Spoiled someone's night out for being too drunk** | 150 | 13.6 | 81 | 19.2 | **0.006** |
| **Failed to attend at university** | 175 | 15.8 | 178 | 42.2 | **<0.001** |
| **Missed exams because of drinking** | 25 | 2.3 | 6 | 1.4 | 0.298 |
| **Missed work because of drinking** | 50 | 4.5 | 35 | 8.3 | **0.004** |

**Table 6. Pre-drinking behaviour as a risk factor for alcohol-related harms after attending nightclubs, bars, and pubs amongst Brazilian and British university students.**

| | After going to nightclubs, bars, and pubs: | | | | | |
| --- | --- | --- | --- | --- | --- | --- |
| | In Brazil | | | In England | | |
| | N = 1,105 | | | N = 422 | | |
| | OR | 95% CI | *p* value | OR | 95% CI | *p* value |
| **Experienced any kind of road traffic accident** | 3.65 | [1.66, 7.99] | **0.001** | * | * | * |
| **Experienced any kind of physical violence** | 2.42 | [1.40, 4.17] | **0.001** | 1.72 | [0.55, 5.38] | 0.345 |
| **Experienced any kind of sexual harassment** | 1.38 | [0.98, 1.93] | 0.060 | 1.84 | [0.74, 4.54] | 0.186 |
| **Had unprotected sex** | 1.91 | [1.36, 2.70] | **<0.001** | 2.29 | [0.82, 6.34] | 0.110 |
| **Regretted a decision to engage in sexual activity** | 3.01 | [1.93, 4.69] | **<0.001** | 3.56 | [0.78, 16.16] | 0.099 |
| **Experienced blackouts, vomiting or coma** | 2.18 | [1.69, 2.80] | **<0.001** | 3.86 | [1.98, 7.52] | **<0.001** |
| **Fallen asleep somewhere inappropriate** | 2.30 | [1.58, 3.33] | **<0.001** | 1.53 | [0.49, 4.76] | 0.457 |
| **Woke up feeling embarrassed** | 1.84 | [1.42, 2.39] | **<0.001** | 2.02 | [1.07, 3.82] | **0.030** |
| **Were refused entry to a nightclub, bar, or pub** | 7.07 | [2.04, 24.48] | **0.002** | 2.54 | [0.82, 7.90] | 0.106 |
| **Spoiled someone's night out for being too drunk** | 1.94 | [1.36, 2.78] | **<0.001** | 1.57 | [0.64, 3.86] | 0.324 |
| **Failed to attend at university** | 2.91 | [2.05, 4.14] | **<0.001** | 4.17 | [1.77, 9.82] | **0.001** |
| **Missed exams because of drinking** | 1.99 | [0.85, 4.68] | 0.111 | * | * | * |
| **Missed work because of drinking** | 2.38 | [1.29, 4.41] | **0.005** | 7.26 | [0.90, 58.11] | 0.061 |

Note:

(*) due to low numbers it was impossible to do meaningful calculations. Regressions controlled by sociodemographic variables (age, gender, marital status, ethnicity, academic year, and alcohol consumption within nightclubs, bars, and pubs settings).

## Discussion

The main purpose of this study was to investigate university students' experiences of pre-drinking and alcohol consumption within nightlife settings, and associated harms in Brazil and England. This is the first study of its kind to compare students' pre-drinking practices in

countries with different drinking guidelines and cultures. The findings contribute to current understanding of pre-drinking behaviour amongst students in a variety of ways, by explaining why students choose to drink before going out and the risks associated whilst doing so. Importantly, comparing two countries with different cultures has revealed how students from Brazil and England have different nightlife experiences and pre-drinking patterns.

Current findings corroborate previous nightlife research conducted in the UK [13, 34] and in Brazil [24] suggesting pre-drinking is a common and socially accepted behaviour among students, often associated with saving money and socializing with peers and usually occurring within unregulated environments, including at home or at a friends' house; and associated with higher levels of drunkenness and alcohol-related harms. However, significant differences in the prevalence of pre-drinking and alcohol consumption between the country samples were found. Thus, while more British students reported pre-drinking, more alcohol was reportedly consumed by Brazilian students during pre-drinking practices. Higher levels of alcohol consumption whilst at nightclubs, bars and pubs were also found amongst Brazilian students when compared with British students. Such findings fill one of the gaps on Brazilian harmful drinking research about students' attitudes towards alcohol consumption within nightlife settings, yet more research is needed to explore pre-drinking behaviour and students' alcohol consumption to develop an understanding of how risk is related to the many ways that students drink alcohol, particularly during pre-drinking [35]. However, it is important to highlight that pre-drinking behaviour amongst students is a complex issue, as it goes beyond the economic factor i.e., students also pre-drink to socialize, which can be a challenge when developing policy and implementing interventions since those aimed at price increase would only change behaviour of a few [21, 36]. This highlights the need for further research exploring the meaningful cultural and social aspects related to students pre-drinking practice.

Corroborating previous research [24, 37], this study found significant associations between pre-drinking and risks of certain alcohol-related harms in nightlife, and this was stronger for British students when compared to Brazilian students. For instance, it was observed that even though Brazilian pre-drinkers reported drinking higher amounts of alcohol during pre-drinking, they reported experiencing less harms from acute intoxication (e.g., blackouts, vomiting or coma) when compared with British students, who were more at risk for experiencing such harms. This could be explained by the difference between the two countries on the culture of alcohol use during a night out. Previous research conducted in the UK suggest that university students have positive views of getting drunk and experiencing alcohol effects such as vomiting or passing out it as a way of entertainment [38, 39], including during pre-drinking practice [40, 41]. Brazilian students who pre-drink might have different drinking patterns when compared with British students, e.g., current results showed a greater proportion of Brazilian students reporting eating food during pre-drinking compared with British students, so eating food whilst drinking might be a strategy that Brazilian students use to avoid experiencing drunkenness. In the UK, a previous study conducted amongst university students [42] showed that eating was considered a problem by the students as it makes alcohol consumption difficult since your stomach was full. Unfortunately, little is known about how people, particularly students control their drinking, highlighting the need to identify and understand how the individual and wider factors influencing students' drinking during a night out vary across cultures.

It is noteworthy that in Brazil, no significant association was observed in the current study between pre-drinkers and reporting any kind of sexual harassment, which did not reflect previous Brazilian findings [23]. Yet, amongst Brazilian pre-drinkers' alcohol might have a greater influence on their perception of risk, by lowering inhibitions and increasing confidence (when compared with British pre-drinkers) as significant association was seen between Brazilian pre-drinkers and increased risks of reporting risky sexual behaviours (e.g., practicing unprotected

sex and regretting a decision to engage in sexual activity), which corroborate previous nightlife research conducted in Brazil [24].

Literature shows that culture is one of the factors influencing drinking patterns [43, 44]. To develop effective measures aimed at reducing pre-drinking and its related harms it is important to investigate how drinking culture varies across countries, however, comparing drinking patterns within nightlife settings between countries is difficult because it can be also influenced by the policy on alcohol consumption of each country [45]. The UK for example has been trying to tackle problems associated with harmful use of alcohol within nightlife settings, including during pre-drinking practice by developing and implementing specific alcohol policies [46, 47] that differ from the laws applied in Brazil. These policies are mainly designated to reduce alcohol affordability (e.g., through implementing higher prices and taxes), availability (e.g., through implementing licenses and restrictions on alcohol sales and outlets) and to restrict alcohol marketing and advertising [48]. Unfortunately, in Brazil, there are no licenses and restrictions for alcohol-selling venues [22], no laws to control closing hours for nightlife establishments, no efficient control on alcohol advertising [49] and it is legal to serve alcohol to drunk people [50]. Equally, much of the research aimed at reducing risks associated with nightlife settings has been conducted in high-income countries [51–54] leaving a gap in low-middle income countries such as Brazil.

When considering students' pre-drinking practice, the availability of cheap alcohol has a strong influence on their drinking pattern [8]. In Brazil and England alcohol can be sold cheaply not only in off-licensed nightlife settings, but also at on-licensed ones during promotional nights in which students are often attracted with the intention to get drunk [22, 55]. Therefore, effective measures are clearly required to reduce students' drunkenness and pre-drinking behaviour, such as banning alcohol discounts prices and promotions, including open bar scheme in Brazil and combo discounts, so that students would not have access to cheap alcohol [22]. However, the current results suggest that amongst students, financial motives seem to be important for motivating pre-drinking. So developing an appropriate policy intervention focused on economic influences can be challenging when considering pre-drinking practices, since measures that increase on-premises prices, without addressing off-premises prices, may favour the consumption of cheap alcohol before attending on-licensed premises, i.e., may aggravate students' pre-drinking practice [24, 56, 57].

Furthermore, to choose the appropriate intervention it is important to consider acceptability amongst the population and investigate its effectiveness. For example, in Brazil, most of the population support increasing taxes on alcoholic drinks [57], limiting hours and places for alcohol sales [58] and restricting alcohol advertising on TV [59]. But, in Brazil, nightlife patrons rejected the idea of imposing law-controls on alcohol sales to drunk people, as this seems to be part of the Brazilian nightlife culture (as opposed to the UK where it is illegal to sell alcohol to drunk people) [60].

This study has some limitations that need to be addressed. It is important to highlight that there are many differences between Brazil and England regarding type of drinks, alcohol strengths and serving sizes (e.g., shots, glasses, pints, and bottle). As opposed to England, where is adopted the concept of counting alcohol units [33], in Brazil there isn't an official definition on how to count alcoholic drinks. Whilst our models controlled for quantity of alcohol consumed within nightlife premises, we did not control for total alcohol consumption across the night out (however pre-drinkers consumed more alcohol overall than non-pre-drinkers). Given the cross-sectional design of this study and the cultural differences between the two countries, results must be interpreted with caution, since causal relations between variables cannot be established and results may not be generalizable to England and Brazil as a whole nor representative of all British and Brazilian university students in general. Despite

limitations, findings from this study resonate with previous studies finding high levels of alcohol consumption during pre-drinking behaviour and its associated risks, and reinforces the importance of local authorities, government, and venues owners cooperating to reduce harmful drinking and its harms, by developing and implementing efficient and well-accepted policies for each one of the countries.

Overall, pre-drinking is a problem amongst younger students in both countries, and it was associated with alcohol-related harms. From a harm reduction perspective, future research should look further into pre-drinking behaviour practice in Brazil and in England, to investigate such practice as part of students' drinking culture. The findings also showed that alcohol policies and interventions within nightlife contexts are important areas for practice and future research.

## Supporting information

**S1 Data.**
(SAV)

## Author Contributions

**Conceptualization:** Mariana G. R. Santos, Zila M. Sanchez, Zara Quigg.

**Formal analysis:** Mariana G. R. Santos.

**Methodology:** Mariana G. R. Santos, Karen Hughes, Ivan Gee, Zara Quigg.

**Project administration:** Mariana G. R. Santos.

**Supervision:** Zila M. Sanchez, Karen Hughes, Ivan Gee, Zara Quigg.

**Validation:** Mariana G. R. Santos, Zila M. Sanchez, Zara Quigg.

**Visualization:** Mariana G. R. Santos.

**Writing – original draft:** Mariana G. R. Santos.

**Writing – review & editing:** Mariana G. R. Santos, Zila M. Sanchez, Karen Hughes, Ivan Gee, Zara Quigg.

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
