## [Decision Letter · Decision Letter 0]

6 Dec 2021

PONE-D-21-35698Pre-drinking, alcohol consumption and related harms amongst Brazilian and British university studentsPLOS ONE

Dear Dr. Quigg,

Thank you for submitting your manuscript to PLOS ONE. After careful consideration, we feel that it has merit but does not fully meet PLOS ONE’s publication criteria as it currently stands. Therefore, we invite you to submit a revised version of the manuscript that addresses the points raised during the review process. The revised version should address all comments. Please submit your revised manuscript by Jan 17 2022 11:59PM. If you will need more time than this to complete your revisions, please reply to this message or contact the journal office at plosone@plos.org. Please include the following items when submitting your revised manuscript:A rebuttal letter that responds to each point raised by the academic editor and reviewer(s). You should upload this letter as a separate file labeled 'Response to Reviewers'.A marked-up copy of your manuscript that highlights changes made to the original version. You should upload this as a separate file labeled 'Revised Manuscript with Track Changes'.An unmarked version of your revised paper without tracked changes. You should upload this as a separate file labeled 'Manuscript'.

We look forward to receiving your revised manuscript.

Kind regards,

Petri Böckerman

Academic Editor

PLOS ONE

Journal Requirements:

" ext-link-type="uri" xlink:type="simple">https://journals.plos.org/plosone/s/file?id=ba62/PLOSOne_formatting_sample_title_authors_affiliations.pdf"

Reviewers' comments:

Reviewer's Responses to Questions

**Comments to the Author**

1. Is the manuscript technically sound, and do the data support the conclusions?

Reviewer #1: Yes

Reviewer #2: Yes

2. Has the statistical analysis been performed appropriately and rigorously? 

Reviewer #1: Yes

Reviewer #2: Yes

3. Have the authors made all data underlying the findings in their manuscript fully available?

Reviewer #1: No

Reviewer #2: No

4. Is the manuscript presented in an intelligible fashion and written in standard English?

Reviewer #1: Yes

Reviewer #2: Yes

5. Review Comments to the Author

Reviewer #1: In the present manuscript the Authors examined Brazilian and British university students’ pre-drinking patterns and associations with nightlife-related harms. The methodological approach, data analysis and conclusions are appropriate. However, there are some concerns and comments that the authors need to address.

1. Among the students you approached via e-mail some were volunteers and the others were abstainers. In your study the abstainers were screened out (excluded). Don’t you think this introduce selection bias? Basically, these two groups may be different in their pre-drinking and other behaviors. How did you handle it?

2. To assess pre-drinking behavior as a risk factor for alcohol-related harms, authors need to consider some very important factors. Because pre-drinking has been linked to subsequent heavy drinking and the engagement in multiple risky behaviors. Factors such as, total amount of alcohol consumption, strengths of the alcohol, frequency of the consumption and the difference in type of drinks in Brazil and the UK were not well described.

3. Minor editorial problems: e.g. . 59.4% of Brazilian students…. (Page 7) …. 80.4% of Brazilian students… (Page 9). Please write numbers and percentage after period in words.

4. No line number. So difficult to indicate the exact paragraph and line where comments found.

Reviewer #2: This study examines predictors and characteristics of pre-drinking, and the impact of pre-drinking on alcohol-related harm, utilizing cross-sectional data from a sample of university students in Brazil and England. Overall, the paper is well-written and provides novel findings regarding cross-cultural differences in pre-drinking and harm prevalence. However, there are some points for improvement:

1. Was the survey validated for use among a Brazilian population? If not, this may be a particular limitation of the study that could be noted.

2. In the regression models predicting alcohol-related harm, the authors account for total alcohol consumption consumed at on-premise venues (i.e., pubs, clubs, etc.), but not total evening consumption, i.e., including drinks consumed off-premise (e.g., at home, friends’ places, etc.). Consequently, the estimates may inadequately control for total consumption, especially for students who consume substantial quantities off-premise. While I’m not familiar with the literature on Brazil, in England, studies show a shift towards off-premise consumption (see below). The authors may wish to note this as a potential limitation.

Davies, E. L., Cooke, R., Maier, L. J., Winstock, A. R., Ferris, J. A. (2021). Where and what you drink is linked to how much you drink: an exploratory survey of alcohol use in 17 countries. Substance Use Misuse, 56(13), 1941-1950.

Meier, P. S. (2010). Polarized drinking patterns and alcohol deregulation: trends in alcohol consumption, harms and policy: United Kingdom 1990–2010. Nordic Studies on Alcohol and Drugs, 27(5), 383-408.

3. In the discussion, the authors note that Brazilian students experienced “less drunkenness effects” compared to British students. Do the authors mean that Brazilian students reported less harms from acute intoxication (e.g., vomiting)? This could be stated more clearly

4. The authors discuss measures for reducing drunkenness and pre-drinking behaviours and suggest banning alcohol discounts and prices, including in bars. Given the price differential between on- and off-premise alcohol appears to be a motivating factor for pre-drinking, strategies that increase on-premise price, without addressing off-premise price may exacerbate pre-drinking. What are the authors thoughts about the unintended consequences of on-premise pricing policies on pre-drinking? (e.g., by making pre-drinking more favourable).

5. A Minor point: In-text citation is inconsistent – both author-date and numbered approaches are used.

6. PLOS authors have the option to publish the peer review history of their article (what does this mean?). If published, this will include your full peer review and any attached files.

Reviewer #1: **Yes: **Agize Asfaw

Reviewer #2: No

---

## [Author Response · Author response to Decision Letter 0]

2 Feb 2022

Journal Requirements:

We ensured that the manuscript follows Plos One style requirement.

2. We note that you have indicated that data from this study are available upon request. PLOS only allows data to be available upon request if there are legal or ethical restrictions on sharing data publicly.

b) b) If there are no restrictions, please upload the minimal anonymized data set necessary to replicate your study findings as either Supporting Information files or to a stable, public repository and provide us with the relevant URLs, DOIs, or accession numbers. 

The minimal anonymized dataset has been uploaded (Supporting Information File S1).

The corresponding author (ZQ) ORCID was validated in Editorial Manager

The reference list has been revised and it is complete and correct. 

Reviewers' comments:

Reviewer #1: 

In the present manuscript the Authors examined Brazilian and British university students’ pre-drinking patterns and associations with nightlife-related harms. The methodological approach, data analysis and conclusions are appropriate. However, there are some concerns and comments that the authors need to address.

1. Among the students you approached via e-mail some were volunteers and the others were abstainers. In your study the abstainers were screened out (excluded). Don’t you think this introduce selection bias? Basically, these two groups may be different in their pre-drinking and other behaviors. How did you handle it?

The survey did not attempt to recruit representative samples from each country but rather samples of alcohol consuming nightlife users in order to explore behaviours amongst such individuals and associations with pre-loading. The method of approach was the same for all participants but non-drinkers were screened out as they were not our intended study population and the questions were not relevant to them. Our aim was not to evaluate prevalence of pre-drinking amongst all students but rather patterns of pre-drinking amongst drinkers and associations between pre-drinking and nightlife harms in alcohol consumers. We have clarified this in the methods throughout the paper by specifying “amongst drinkers” (in lines 30, 169, 240)

2. To assess pre-drinking behavior as a risk factor for alcohol-related harms, authors need to consider some very important factors. Because pre-drinking has been linked to subsequent heavy drinking and the engagement in multiple risky behaviors. Factors such as, total amount of alcohol consumption, strengths of the alcohol, frequency of the consumption and the difference in type of drinks in Brazil and the UK were not well described.

We agree with reviewer’s point. 

Regarding how the total amount of alcohol consumption variable was created we added the following information (described in lines 141-157): “Brazil and England have many differences regarding type of drinks and glass/bottle sizes, and because the Brazilian government does not have an official definition of how to count alcoholic drinks, the variables related to quantity of alcohol were recoded using the UK definition, according to which one unit is equal to 10 ml or 8 g of pure alcohol. To calculate the alcohol amount consumed between countries, drinks were coded into standard UK units by multiplying the total volume of an alcoholic drink (ml) by its alcohol content (using its ABV measure – alcohol by volume) and dividing the result by 1,000 (Table 1).”

The total quantity of alcohol consumed during a night out variable was created by the sum of total amount of each type of drinks, such as: total amount of spirits (e.g., those who reported consuming bottle, single or double measures were grouped); total amount of wine (e.g., those who reported consuming bottle, small, standard, or large glasses were grouped); total amount of beer (e.g., those who reported consuming bottle, can, pin or half pint were grouped); and, total amount of alcopop (e.g., those who reported consuming large, standard or can were grouped).

Also, we describe in lines 199-204, that “Regarding the median number of total alcohol units (i.e., reported drinking any alcohol), amongst pre-drinkers, Brazilian students reported drinking a median of 17.5 units of alcohol compared with 12.1 units for British students U=70996.0, p0.001). Amongst non-pre-drinkers, Brazilian students reported drinking a median of 16.6 units of alcohol on on-licensed premises compared with 8.2 units for British students (U=13317.5, p0.001).”

3. Minor editorial problems: e.g. . 59.4% of Brazilian students…. (Page 7) …. 80.4% of Brazilian students… (Page 9). Please write numbers and percentage after period in words.

We corrected the sentences to: “Regarding marital status, 59.4% of Brazilian students…” (line 180) and “Regarding mixing food with drinking during pre-drinking, 80.4% of Brazilian students reported…” (lines 197-198).

4. No line number. So difficult to indicate the exact paragraph and line where comments found.

We have now added line numbers throughout the manuscript.

Reviewer #2: 

This study examines predictors and characteristics of pre-drinking, and the impact of pre-drinking on alcohol-related harm, utilizing cross-sectional data from a sample of university students in Brazil and England. Overall, the paper is well-written and provides novel findings regarding cross-cultural differences in pre-drinking and harm prevalence. However, there are some points for improvement:

1. Was the survey validated for use among a Brazilian population? If not, this may be a particular limitation of the study that could be noted.

It was added the following information (described in lines 113-115): “A convenience sample of English and Brazilian students was recruited through informal networks to test the developed questionnaire. This pilot study was conducted in February 2017”.

2. In the regression models predicting alcohol-related harm, the authors account for total alcohol consumption consumed at on-premise venues (i.e., pubs, clubs, etc.), but not total evening consumption, i.e., including drinks consumed off-premise (e.g., at home, friends’ places, etc.). Consequently, the estimates may inadequately control for total consumption, especially for students who consume substantial quantities off-premise. While I’m not familiar with the literature on Brazil, in England, studies show a shift towards off-premise consumption (see below). The authors may wish to note this as a potential limitation.

Davies, E. L., Cooke, R., Maier, L. J., Winstock, A. R., Ferris, J. A. (2021). Where and what you drink is linked to how much you drink: an exploratory survey of alcohol use in 17 countries. Substance Use Misuse, 56(13), 1941-1950.

Meier, P. S. (2010). Polarized drinking patterns and alcohol deregulation: trends in alcohol consumption, harms and policy: United Kingdom 1990–2010. Nordic Studies on Alcohol and Drugs, 27(5), 383-408.

We agree with reviewer’s point. 

At the manuscript, we describe (in lines 172-175), that “The quantity of alcohol consumption in nightlife settings was included in the model as a control variable because the analysis aimed to investigate how pre-drinking behaviour would affect alcohol-related harms, independent of alcohol consumption occurring in nightlife settings”. 

Also, as I previously mentioned in reviewer’s 1 comment, it was added the following information described in lines 199-204, that “Regarding the median number of total alcohol units (i.e., reported drinking any alcohol), amongst pre-drinkers, Brazilian students reported drinking a median of 17.5 units of alcohol compared with 12.1 units for British students U=70996.0, p0.001). Amongst non-pre-drinkers, Brazilian students reported drinking a median of 16.6 units of alcohol on on-licensed premises compared with 8.2 units for British students (U=13317.5, p0.001)”. Whilst our models controlled for quantity of alcohol consumed within nighlife premises, we did not control for total alcohol consumption across the night out because these two variables were closely related to include in the same model. 

It was also included as a limitation (described in lines 353-363): “It is important to highlight that there are many differences between Brazil and England regarding type of drinks, alcohol strengths and serving sizes (e.g., shots, glasses, pints, and bottle). As opposed to England, where is adopted the concept of counting alcohol units, in Brazil there isn’t an official definition on how to count alcoholic drinks. Whilst our models controlled for quantity of alcohol consumed within nightlife premises, we did not control for total alcohol consumption across the night out (however pre-drinkers consumed more alcohol overall than non-pre-drinkers). Given the cross-sectional design of this study and the cultural differences between the two countries, results must be interpreted with caution, since causal relations between variables cannot be established and results may not be generalizable to England and Brazil as a whole nor representative of all British and Brazilian university students in general.”

3. In the discussion, the authors note that Brazilian students experienced “less drunkenness effects” compared to British students. Do the authors mean that Brazilian students reported less harms from acute intoxication (e.g., vomiting)? This could be stated more clearly

We changed the sentence to: “…they reported experiencing less harms from acute intoxication (e.g., blackouts, vomiting or coma) when compared with British students, who were more at risk for experiencing such harms” (lines 293-295).

4. The authors discuss measures for reducing drunkenness and pre-drinking behaviours and suggest banning alcohol discounts and prices, including in bars. Given the price differential between on- and off-premise alcohol appears to be a motivating factor for pre-drinking, strategies that increase on-premise price, without addressing off-premise price may exacerbate pre-drinking. What are the authors thoughts about the unintended consequences of on-premise pricing policies on pre-drinking? (e.g., by making pre-drinking more favourable).

In another paper under preparation, we will present findings from this study, but regarding students’ perceived impact that alcohol policy measures would have on their pre-drinking behaviour. We found differences amongst Brazilian and British students regarding their perceived impact of increasing alcohol prices (in both on and off-licensed premises) in pre-drinking practice, which highlights the importance of economic influence. 

We added the following information (described in lines 340-345): “the current results suggest that amongst students, financial motives seem to be important for motivating pre-drinking. So developing an appropriate policy intervention focused on economic influences can be challenging when considering pre-drinking practices, since measures that increase on-premises prices, without addressing off-premises prices, may favour the consumption of cheap alcohol before attending on-licensed premises, i.e., may aggravate students’ pre-drinking practice.”

5. A Minor point: In-text citation is inconsistent – both author-date and numbered approaches are used.

In-text citation was revised and corrected.

---

## [Decision Letter · Decision Letter 1]

18 Feb 2022

Pre-drinking, alcohol consumption and related harms amongst Brazilian and British university students

PONE-D-21-35698R1

Dear Dr. Quigg,

We’re pleased to inform you that your manuscript has been judged scientifically suitable for publication and will be formally accepted for publication once it meets all outstanding technical requirements.

Kind regards,

Petri Böckerman

Academic Editor

PLOS ONE

Additional Editor Comments (optional):

Reviewers' comments:

Reviewer's Responses to Questions

**Comments to the Author**

1. If the authors have adequately addressed your comments raised in a previous round of review and you feel that this manuscript is now acceptable for publication, you may indicate that here to bypass the “Comments to the Author” section, enter your conflict of interest statement in the “Confidential to Editor” section, and submit your "Accept" recommendation.

Reviewer #2: All comments have been addressed

2. Is the manuscript technically sound, and do the data support the conclusions?

Reviewer #2: Yes

3. Has the statistical analysis been performed appropriately and rigorously? 

Reviewer #2: Yes

4. Have the authors made all data underlying the findings in their manuscript fully available?

Reviewer #2: Yes

5. Is the manuscript presented in an intelligible fashion and written in standard English?

Reviewer #2: Yes

6. Review Comments to the Author

Reviewer #2: Thank you for the thoughtful and considered responses. I look forward to reading the forthcoming study on student perceptions of alcohol policy measures. The comments and concerns that I raised have now been adequately addressed. Well done!

7. PLOS authors have the option to publish the peer review history of their article (what does this mean?). If published, this will include your full peer review and any attached files.

Reviewer #2: No

---

## [Editor Report · Acceptance letter]

8 Mar 2022

PONE-D-21-35698R1 

Pre-drinking, alcohol consumption and related harms amongst Brazilian and British university students 

Dear Dr. Quigg:

I'm pleased to inform you that your manuscript has been deemed suitable for publication in PLOS ONE. Congratulations! Your manuscript is now with our production department. 

Kind regards, 

on behalf of

Professor Petri Böckerman 

Academic Editor

PLOS ONE